# Extra kinetic dimensions for label discrimination

Raja Chouket[1,9], Agnès Pellissier-Tanon [1,9], Aliénor Lahlou [1,2], Ruikang Zhang [1], Diana Kim [1], Marie-Aude Plamont[1], Mingshu Zhang [3], Xi Zhang[3], Pingyong Xu [3,4], Nicolas Desprat[5,6], Dominique Bourgeois [7], Agathe Espagne[1], Annie Lemarchand [8✉], Thomas Le Saux [1✉] & Ludovic Jullien [1✉]

Due to its sensitivity and versatility, fluorescence is widely used to detect specifically labeled biomolecules. However, fluorescence is currently limited by label discrimination, which suffers from the broad full width of the absorption/emission bands and the narrow lifetime distribution of the bright fluorophores. We overcome this limitation by introducing extra kinetic dimensions through illuminations of reversibly photoswitchable fluorophores (RSFs) at different light intensities. In this expanded space, each RSF is characterized by a chromatic aberration-free kinetic fingerprint of photochemical reactivity, which can be recovered with limited hardware, excellent photon budget, and minimal data processing. This fingerprint was used to identify and discriminate up to 20 among 22 spectrally similar reversibly photoswitchable fluorescent proteins (RSFPs) in less than 1s. This strategy opens promising perspectives for expanding the multiplexing capabilities of fluorescence imaging.

[1] PASTEUR, Département de chimie, École normale supérieure, PSL University, Sorbonne Université, CNRS, Paris, France. [2] Sony Computer Science Laboratories, Paris, France. [3] Key Laboratory of RNA Biology, Institute of Biophysics, Chinese Academy of Sciences, Beijing, China. [4] College of Life Sciences, University of Chinese Academy of Sciences, Beijing, China. [5] Laboratoire de Physique de l'ENS, École Normale Supérieure, PSL University, CNRS, Sorbonne Université, Université de Paris, Paris, France. [6] Institut de Biologie de l'ENS (IBENS), École Normale Supérieure, CNRS, INSERM, PSL University, Paris, France. [7] Univ. Grenoble Alpes, CNRS, CEA, IBS, F-38000 Grenoble, France. [8] LPTMC, Sorbonne Université, CNRS, Paris, France. [9] These authors contributed equally: Raja Chouket, Agnès Pellissier-Tanon. ✉email: Annie.Lemarchand@sorbonne-universite.fr; Thomas.Lesaux@ens.psl.eu; Ludovic.Jullien@ens.psl.eu

There is a growing demand in quantitative biology to simultaneously image tens of different molecules[1]. Hence powerful genetic engineering strategies have been introduced for labeling biomolecules or cells[2–4] but multiplexing is presently limited by label discrimination[5]. 3-4 fluorophores can be spectrally discriminated in real-time fluorescence imaging, which is valued for its high sensitivity and widespread availability. Advanced data processing can further increase this number to 7-9 but at the cost of a degraded photon budget and an increased computation time[6–8]. In this report, we introduce Light-tunable tIme-gated readinG-out of pHotocycles for mulTiplexed fluorescence ImagiNG (LIGHTNING), which is promising to boost this number.

The simplest absorption–fluorescence emission photocycle is usually exploited to target a label. It provides two spectral – the absorption and emission spectra—and one temporal—the fluorescence lifetime—discriminative dimensions (see Fig. 1a). Yet, its discriminative power is intrinsically limited by the broad full width of the absorption/emission bands and the narrow lifetime dispersion of the bright fluorophores. Endowed with several photoactivatable states and a wide palette of relaxation times associated with photochemical and thermal steps[9], the photocycles of reversibly photoswitchable fluorophores (RSFs) open many more dimensions, which have only been partially used for label discrimination. State-of-the-art imaging protocols of dynamic contrast usually exploit a single temporal dimension[10–13], so that distinguishing many RSFs requires a large span of characteristic times and a large time window for signal acquisition. The protocols involving an oscillating light excitation based on matching the period of the excitation and a typical time scale of photoswitching dynamics[13,14] further necessitate a kinetic model of RSF photoswitching and successive acquisitions at optimal frequency targeting individual RSFs. By uncovering new kinetic dimensions, LIGHTNING considerably increases the number of RSFs which can be identified and distinguished without specific information on their detailed photoswitching mechanism and by applying a single illumination sequence common to all RSF labels, which considerably shortens image acquisition.

## Results

**The LIGHTNING concept.** In LIGHTNING, the fluorescent labels are RSFs, which are engaged in first-order light- and thermally-driven reactions (see Fig. 1b). Upon photoactivation, the concentration of the RSF states which exhibit different brightnesses, and the RSF fluorescence signal evolve as a linear combination of exponential terms (see section A in Supplementary Information). The amplitudes and relaxation times of these exponential terms depend on the multiple rate constants of the photocycle reactions. The light intensity affects the rate constants of the photochemical steps but not those of the thermal steps. We exploit this crucial property to change the rate-limiting steps of the photocycle, which enables us to extract rich kinetic content specific to each RSF.

LIGHTNING harnesses the kinetics of fluorescence evolution without any hypothesis on the underlying photochemical network and the number of exponential components. Instead of multiexponential data processing which is poorly robust[15], each evolution of the RSF fluorescence signal is parameterized by a single characteristic time $\tau$ regardless of the photocycle complexity. Applying a sequence of $n$ multicolor illuminations at different light intensities yields a set of at least $n$ non-redundant characteristic times $\{\tau_i\}$, which provides the LIGHTNING kinetic fingerprint (see Fig. 1b). Importantly, a single illumination sequence can be tailored to deliver the LIGHTNING kinetic fingerprints of all non-colocalized RSFs present in the sample. This expansion into new kinetic dimensions takes advantage of photochemical reactivity and enables LIGHTNING to identify and discriminate a large number of RSFs using one of the following simple data processing methods: (i) For sufficiently

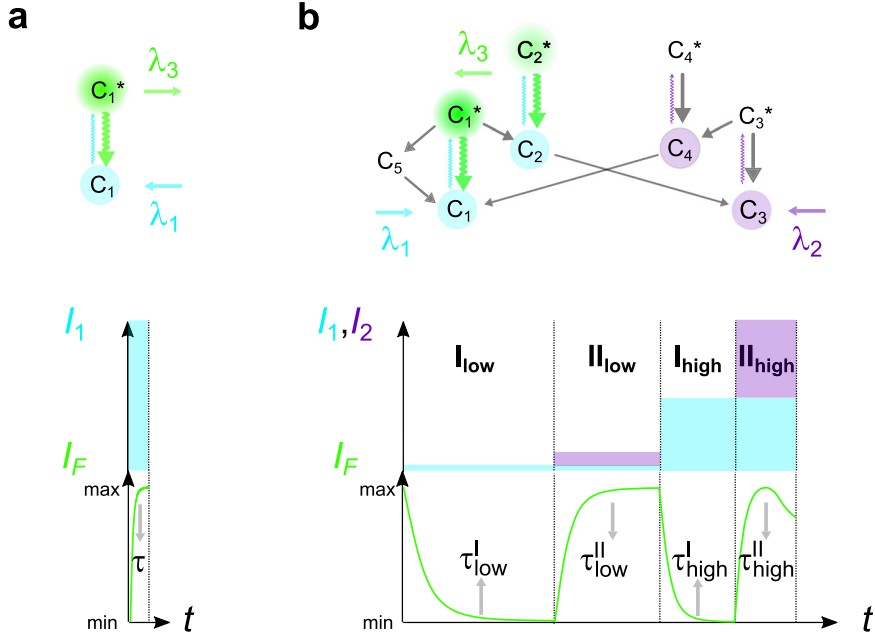

**Fig. 1 LIGHTNING opens new dimensions for discriminating fluorescent labels. a** Illumination of the standard absorption/emission photocycle yields three dimensions to discriminate a label: two wavelengths $\{\lambda_1,\lambda_3\}$ for photoactivation and fluorescence emission, and a lifetime $\tau$; **b** LIGHTNING uses the rich photocycle of RSFs involving several states photoactivatable at distinct wavelengths. Its probing with a multicolored sequence of illuminations at various light intensities generates multiple fluorescence time responses and as many characteristic times. With RSFPs, four illuminations $\{I_{low}, II_{low}, I_{high}, II_{high}\}$ activating the photocycle at different light intensities at $\lambda_1 = 488$ (blue) and $\lambda_2 = 405$ (purple) nm and fluorescence readout at $\lambda_3 = 525$ nm (green) give seven dimensions for discrimination: three wavelengths $\{\lambda_1,\lambda_2,\lambda_3\}$, and a kinetic fingerprint of four non-redundant characteristic times $\{\tau^I_{low}, \tau^{II}_{low}, \tau^I_{high}, \tau^{II}_{high}\}$.

close uncertainties on the $\{\tau_i\}$, two RSFs are distinguished when the Euclidean distance between their positions in the $n$-dimensional space ($\{\tau_i\}$) is greater than the resolution limit governed by the experimental uncertainty. Then, the presence of a referenced RSF in a sample is ascertained when the distance between the kinetic fingerprints of the sample and the referenced RSF is smaller than the resolution limit (see section 1.7 in Supplementary Information); (ii) For different uncertainties on the $\{\tau_i\}$, we compute the probability that a sample is identical to each referenced RSF and assign the identity of the referenced RSF with the maximum probability to the sample (see section 1.8 in Supplementary Information).

**LIGHTNING data collection**. In order to explore the LIGHT-NING capability for discrimination, we used reversibly photo-switchable fluorescent proteins (RSFPs)[16], which have been popularized by super-resolution microscopy[9]. We engineered and selected a series of Skylan[17] mutants differing by the $\alpha$-amino acids at positions 62, 157, and 173. These Skylans are endowed with high brightness, high on-off contrast, and various photo-switching kinetics (see section 1 in Supplementary Information). After addition of Dronpa[18] and rsEGFP[19] mutants, we obtained a collection of 22 spectrally similar green RSFPs (**1**–**22**) (see section 1 in Supplementary Information).

We developed a dedicated optical setup to acquire the RSFP kinetic information for LIGHTNING implementation. We recorded the evolution of the fluorescence signal of 10–20 $\mu$M RSFP solutions at $\lambda_3 = 525$ nm upon applying sequences of photoactivating illuminations at $\lambda_1 = 488$ nm only (illuminations I, constant light of intensity $I_1$), or at both $\lambda_1$ and $\lambda_2 = 405$ nm (illuminations II, constant lights of intensities $I_1$ and $I_2$) for tens of light intensities covering 5 orders of magnitude up to $10^4$ W cm$^{-2}$ (see Fig. 2a and Supplementary Figure 21–Supplementary Figure 42a, b). The observed fluorescence evolutions have been processed by adopting a phenomenological approach: we used a monoexponential function to fit to the part of the fluorescence evolution exhibiting the largest amplitude over the time window for signal acquisition in order to retrieve a single characteristic time $\tau^I$ ($\tau^{II}$, resp.) for each illumination I (II, resp.) (see section B in Supplementary

Information). The extracted characteristic time cannot be necessarily interpreted as a physical time of chemical kinetics. Nevertheless, we propose a mechanism of RSFP photocycle (see Fig. 1b) and discuss the illuminations for which the extracted characteristic time matches an intrinsic property of kinetics (see section C in Supplementary Information).

All investigated RSFPs share a qualitatively similar light intensity dependence of the characteristic times (see section C in Supplementary Information). $1/\tau^I$ ($1/\tau^{II}$, resp.) linearly increases with $I_1$ ($I_2$, resp.) for sufficiently low intensities and tends to saturate for higher intensities (see Fig. 2b and Supplementary Figure 21–Supplementary Figure 42c–f). At the lowest intensities, the photochemical steps limit the rate of the RSFP photocycle and fluorescence evolution is monoexponential. The photoswitching cross-sections are deduced from the slopes of $1/\tau^I$ versus $I_1$ and $1/\tau^{II}$ versus $I_2$, respectively (see section A in Supplementary Information). At the highest intensities, thermal steps intervene in the photocycle rate, and $\tau^I$ and $\tau^{II}$ exhibit no significant—or a weaker—dependence on light intensities. The low- and high-intensity regimes are delimited by indicative threshold intensities $I_1^c$ in the range 2–80 Ein m$^{-2}$ s$^{-1}$ (50–2000 W cm$^{-2}$) at $\lambda_1$ and $I_2^c$ in the range 0.4–10 Ein m$^{-2}$ s$^{-1}$ (12–300 W cm$^{-2}$) at $\lambda_2$ (see Supplementary Table 3).

**LIGHTNING in action**. Due to their different light intensity dependence, the characteristic times acquired under illuminations I and II in the regimes of low and high intensities are non-redundant for kinetic discrimination (see section C in Supplementary Information). We here retained four illuminations $\{I_{low}, II_{low}, I_{high}, II_{high}\}$ associated with light intensities enabling us to achieve each regime for all RSFPs while minimizing the acquisition duration. We defined the RSFP LIGHTNING kinetic fingerprint as the set of four characteristic times $\{\tau^I_{low}, \tau^{II}_{low}, \tau^I_{high}, \tau^{II}_{high}\}$ obtained for the ordered sequence $\{I_{low}, II_{low}, I_{high}, II_{high}\}$.

We first examined the LIGHTNING discriminatory power in 10 $\mu$M solutions of the 22 RSFPs. As shown in Fig. 3a, b, the similarity of their absorption and emission spectra hinders spectral discrimination. We recorded the evolution of the fluorescence signal at 1 MHz acquisition frequency in the

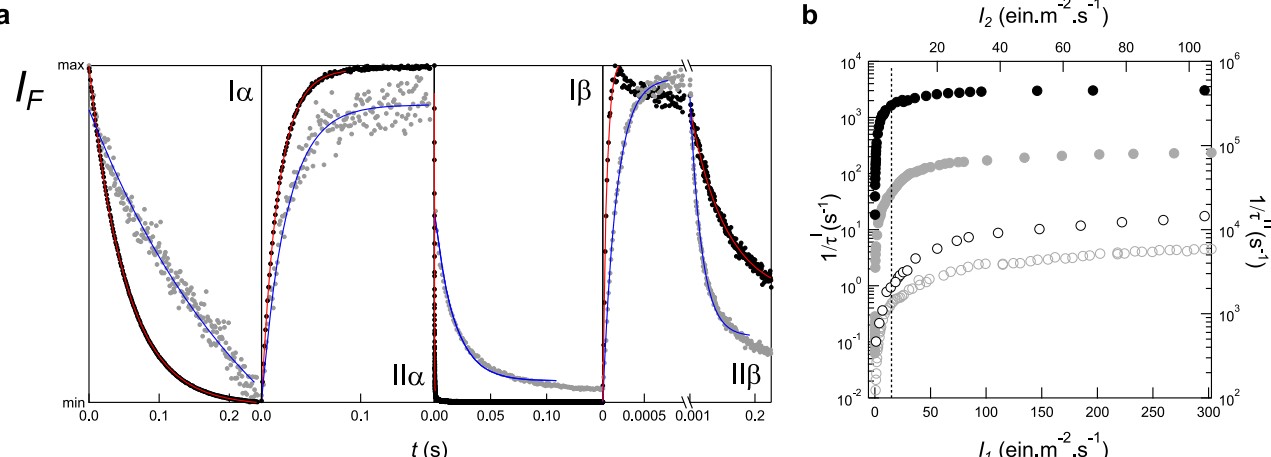

**Fig. 2 LIGHTNING data collection on RSFP solutions. a** Four illuminations involving lights at $\lambda_1 = 488$ and $\lambda_2 = 405$ nm at respective intensities $I_1$ and $I_2$ ($I_1$, $I_2$ in Ein m$^{-2}$ s$^{-1}$; I$\alpha$: $I_1 = 0.1$, II$\alpha$: $I_1 = 0.1$ and $I_2 = 0.1$, I$\beta$: $I_1 = 50$, II$\beta$: $I_1 = 50$ and $I_2 = 20$) yield four fluorescence evolutions $I_F$ (markers; each scaled by the difference between the maximal and minimal values) at $\lambda_3 = 525$ nm. Whereas I$\alpha$, II$\alpha$, and I$\beta$ provide a single characteristic time after monoexponential fitting (lines), up to two characteristic times can be retrieved in II$\beta$ depending on the frequency and time window of fluorescence acquisition; **b** Inverse of the characteristic time obtained for a series of illuminations I (disks) (II (circles), resp.) versus light intensity $I_1$ ($I_2$, resp. at $I_1 = 0.2$ Ein m$^{-2}$ s$^{-1}$). The vertical dotted line separates the low-intensity regime for which the rate-limiting step is photochemical from the high-intensity regime for which the rate-limiting step is thermal. **1** (grey markers and blue lines) and **2** (black markers and red lines) are representatives of slow and fast photoswitching RSFPs respectively.

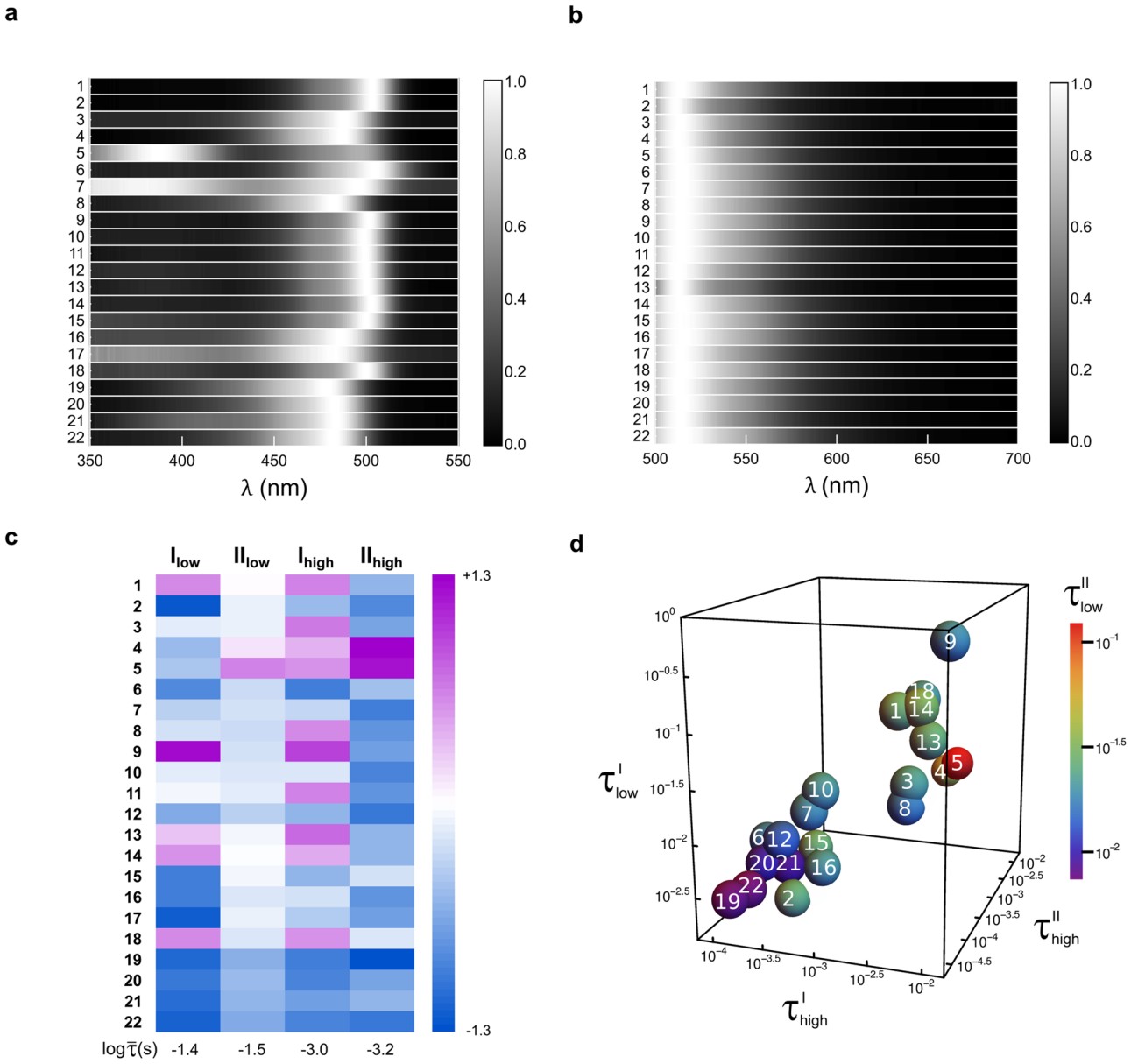

**Fig. 3 LIGHTNING in action in solutions of RSFPs.** Normalized absorption (**a**) and emission (**b**; excitation at $\lambda_1 = 488$ nm) spectra of $5\,\mu$M solutions of the RSFPs **1–22** in pH 7.4 PBS buffer with linear grey scale between 0 and 1; **c** Heatmap displaying the deviations from the mean value $\bar{\tau}$ over the RSFPs of the characteristic times measured in $10\,\mu$M solutions over the $[1\,\mu$s–1 s$]$ time window for each illumination in decimal logarithmic scale; **d** LIGHTNING discrimination in RSFP solutions in the 4D space of characteristic times. The volume associated with each RSFP is fixed by the resolution limit. $\text{I}_{\text{low}}$ ($I_1 = 2$), $\text{II}_{\text{low}}$ ($I_1 = 0.1$, $I_2 = 0.1$), $\text{I}_{\text{high}}$ ($I_1 = 200$), and $\text{II}_{\text{high}}$ ($I_1 = 2$ and $I_2 = 90$) with $I_1$ and $I_2$ in ein m$^{-2}$ s$^{-1}$. $T = 298$ K.

$[1\mu$s–1s$]$ time window (see sections 1.3 and A.2.2 in Supplementary Information). Figure 3c sums up the resulting $\{\tau^{\text{I}}_{\text{low}}, \tau^{\text{II}}_{\text{low}}, \tau^{\text{I}}_{\text{high}}, \tau^{\text{II}}_{\text{high}}\}$ sets, where $\tau^{\text{II}}_{\text{high}}$ characterizes the fast increase of fluorescence (see Supplementary Table 7). Then we determined the optimized subsets of any size containing the most distant RSFPs (see section D in Supplementary Information). In order to evaluate the resolution limit (see section D in Supplementary Information), we measured 500 times the fingerprint of two representative RSFPs, which led us to evaluate the minimum distance between two discriminatable RSFPs at $d_c = 0.20$ (see Eq. (15) in Supplementary Information). Here, LIGHTNING unambiguously identified and distinguished 20 RSFPs among the 22 investigated RSFPs, as evidenced by the spreading and lack of overlap of the fingerprints in the discriminating 4D space (see Fig. 3d and Supplementary Figure 47a).

Then we modified our optical setup to image 16 strains of fixed *Escherichia coli* bacteria each expressing a given RSFP at 1 kHz acquisition frequency over the $[1\,\text{m s–1 s}]$ time window. The images have been processed at the individual bacterium level to extract the distributions of the characteristic times measured from a large number of bacteria. Due to the temporal resolution of 1 ms, the fast fluorescence increase was not accessible under illumination $\text{II}_{\text{high}}$ and the slower decrease of fluorescence has been harnessed. Moreover, in line with the photoswitching cross-sections and light intensities, no significant time evolution of the fluorescence signal from RSFP-labeled bacteria was accessible under illumination $\text{II}_{\text{low}}$. Hence, the LIGHTNING fingerprint was reduced to $\{\tau^{\text{I}}_{\text{low}}, \tau^{\text{I}}_{\text{high}}, \tau^{\text{II}}_{\text{high}}\}$ (see Fig. 4a, Supplementary Figure 43–Supplementary Figure 46, and Supplementary Table 13) without significant loss of discrimination

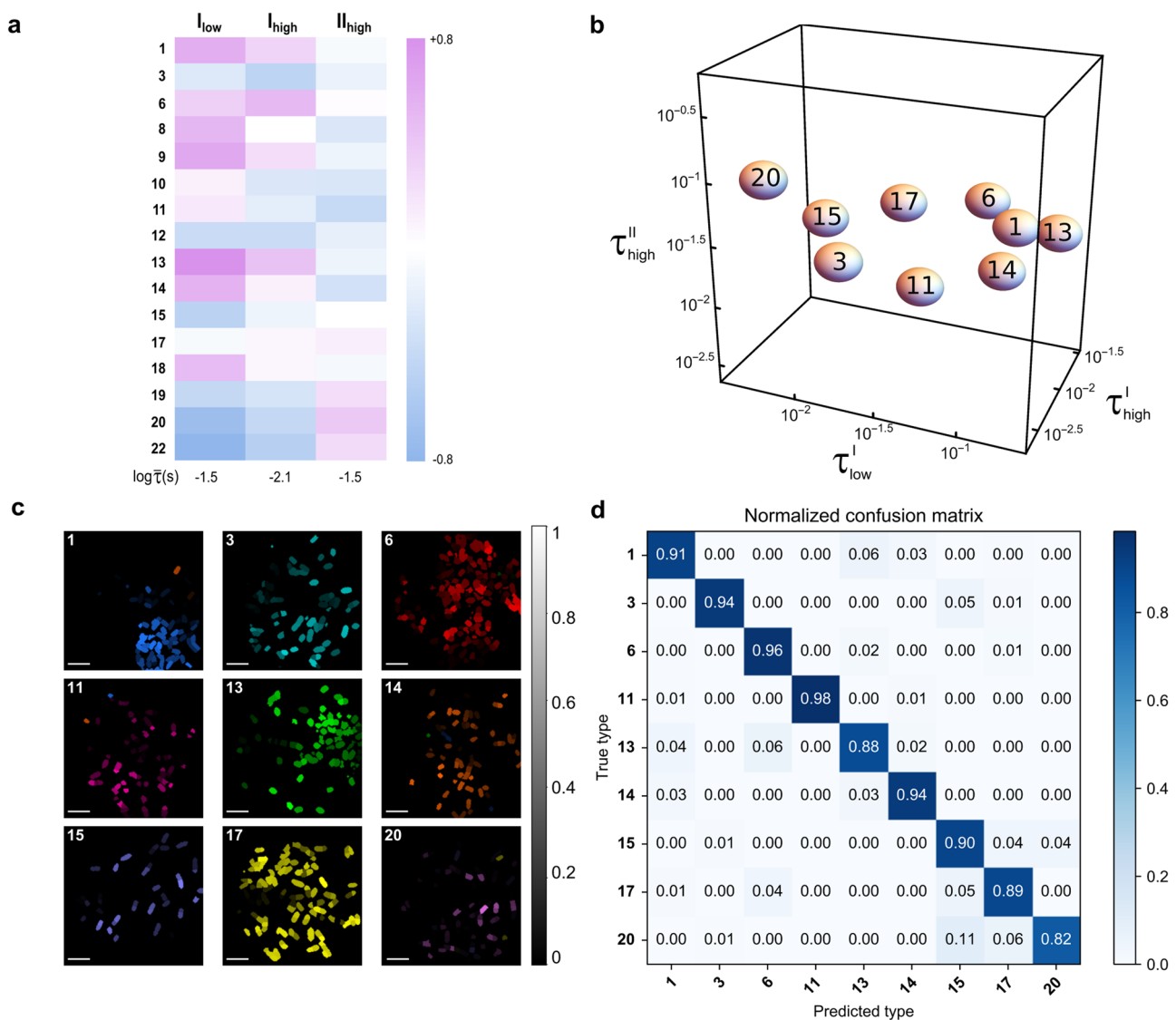

**Fig. 4 LIGHTNING in action in fixed RSFP-expressing *Escherichia coli* bacteria. a** Heatmap displaying the deviations from the mean value $\overline{\tau}$ over the RSFPs of the characteristic times measured over the [1 ms–1 s] time window for each illumination for the RSFP-labeled bacteria in decimal logarithmic scale; **b** LIGHTNING discrimination in RSFP-labeled bacteria in the 3D space of accessible characteristic times. The volume associated with each RSFP is fixed by the resolution limit; **c** LIGHTNING images of 9 types of RSFP-labeled bacteria. The maximum probability that a bacterium is identical to a tabulated RSFP-labeled bacterium sets the lightness (linear grey scale between 0 and 1). A false color is assigned to each predicted type and the true type is indicated in the top left of the image of each sample. Scaling bar = 5 $\mu$m; **d** Confusion matrix associated with the 9 discriminated types of RSFP-labeled bacteria (proportion of true and false positives with linear scale between 0 and 1). $I_{low}$ ($I_1 = 2$), $II_{low}$ ($I_1 = 2$, $I_2 = 0.2$), $I_{high}$ ($I_1 = 50$), and $II_{high}$ ($I_1 = 50$ and $I_2 = 20$) with $I_1$ and $I_2$ in ein m$^{-2}$ s$^{-1}$. T = 298 K.

power due to the narrow dispersion of $\tau_{low}^{II}$ (see Fig. 3c, and Supplementary Table 7). The average values and standard deviations of the distributions of characteristic times were used to compute the resolution limit $d_c = 0.28$ and identify an optimized subset of 9 RSFP-expressing bacteria. They were also exploited to compute the maximum probability that a bacterium is identical to a tabulated RSFP-labeled bacterium, the identity of which was assigned to the bacterium (see sections 1.8 and D.2 in Supplementary Information). Figure 4c illustrates the results and the confusion matrix shown in Fig. 4d gives the proportion of true and false positives. The accuracy defined as the ratio of the number of correctly identified bacteria and the total number of RSFP-labeled bacteria is equal to 0.93, establishing that LIGHT-NING can reliably identify 9 RSFPs-labeled bacteria among 16.

Eventually, we studied the fluorescence response of two representative RSFPs **1** and **2** in three distinct environments – solution, fixed, and living labeled *Escherichia coli* bacteria – to various {$I_{low}$, $II_{low}$, $I_{high}$, $II_{high}$} illuminations using an acquisition frequency of 100 kHz in the [1$\mu$s–0.25 s] time window (see section D.3 in Supplementary Information). For each illumination regime, (i) the dependence of the characteristic times on the light intensities agrees with the prediction of the eight-state photocycle displayed in Fig. 1b (see section C in Supplementary Information); (ii) the three distinct environments lead to similar characteristic times differing by a factor of about 1.2 for **1** and 1.4 for **2** in average, which is in line with the consistency of $\tau_{low}^{I}$ and $\tau_{high}^{I}$ in Figs. 3c and 4a (see section D.3 in Supplementary Information). The changes in environment induce a variability of the characteristic times, which remains lower than the resolution limit of the kinetic fingerprint determined for a large number of acquisitions in a given environment.

## Discussion

LIGHTNING characterizes the kinetics of reversibly photoswitchable fluorophores (RSFs) by non-redundant characteristic times obtained for different illuminations. The green RSFPs have emerged as favorable labels to its implementation: (i) They exhibit a rich kinetics; (ii) They yield a satisfactory photon budget: Most emitted photons are processed to extract the kinetic fingerprint and their number is only reduced by a factor of 5 (5/4, resp.) under illumination I (II, resp.) with regards to non-photoswitchable fluorophores sharing similar photophysical features (see section A in Supplementary Information); (iii) Their kinetic fingerprint has been found robust when comparing three different environments: solution, fixed, and living bacteria, which is in line with previously reported sensitivity of RSFP photoswitching kinetics to environmental changes[13,14,20–22]. Hence, LIGHTNING succeeded in discriminating 20 RSFPs among 22 in solution and 9 fixed RSFP-labeled *Escherichia coli* bacteria among 16 in less than 1 s (see section A in Supplementary Information) at light intensities used in wide-field microscopies (e.g., total internal reflection fluorescence – TIRF– microscopy), which do not introduce any detrimental phototoxicity[23].

These results make LIGHTNING promising for chromatic-aberration-free, live multiplexed fluorescence imaging. Its implementation does not require a rich hardware of light sources, optics corrected for chromatic aberration, dichroic mirrors, and optical filters, nor any demanding data processing for image extraction. Retrieving a LIGHTNING kinetic fingerprint is robust even when the targeted RSF is not the only fluorescent species (see section B in Supplementary Information). When the interfering fluorescence signal is constant (e.g., autofluorescence), it is intrinsically discarded from data processing. When the interfering signal evolves with a characteristic time sufficiently different from the target, it is ignored by using an appropriate time window, which plays the role of a kinetic filter, i.e., selects a targeted characteristic time. Eventually, we developed a protocol to eliminate a background of fluorescence evolution associated with a characteristic time close to the target one (see section D.4 in Supplementary Information). Interestingly, this protocol will be relevant to extract the kinetic fingerprints of colocalized RSFs.

In this work, LIGHTNING has been implemented to successfully discriminate an unprecedented number of fluorescent proteins using simple tools. However, its principle could be transposed to other spectroscopies than fluorescence for label reporting under illumination (e.g., to further improve multiplexing in Raman microscopy[5]). LIGHTNING also promises further improvements. For the sake of simplicity, we characterized the kinetics of an RSFP with a single characteristic time per illumination regime. Yet, two characteristic times can be straightforwardly extracted from the non monotonous fluorescence evolution under illumination $II_{high}$ by applying a robust monoexponential curve fitting in each interval of monotonous evolution, which would add another dimension for RSFP discrimination. With an optimized acquisition sequence, LIGHTNING will be relevant in light scanning (e.g., confocal or light-sheet) microscopies, where RSFPs may benefit from further discriminative dimensions involving new time windows, photo-activating wavelengths, and regimes of light intensity. As in other approaches for multiplexed fluorescence imaging[24,25], it presently exploits a simple fingerprint but machine learning algorithms could further increase its discrimination capabilities. Eventually LIGHTNING could contain information on the concentrations of the RSFs from mining the amplitudes of the fluorescence evolutions.

## Methods

**Development of Skylan serials**. We have previously reported that a serial of RSFPs with versatile properties could be obtained by site-saturation mutagenesis of the first amino acid of the chromophore triplet (XYG) in a photoconvertible fluorescent protein (PCFP)[26]. Using the same strategy, we developed new RSFPs based on the PCFP mEos3.1, which can be denoted as Skylan-X (X represents the α-amino-acid at 62, the first one of the chromophore triplet). Among which two: Skylan-S[27] (herein Skylan-SVF) and Skylan-NS[17] (herein Skylan-LVF) had been reported for their optimal performance in live super-resolution microscopy. Next, to further diversify the photoswitching kinetics of Skylan-X, we did site-directed mutagenesis at 157 and 173, which had been proven to affect the brightness[28] and photoswitching kinetics[16]. All the mutants were then sequenced, and the plasmids were extracted and transformed into Escherichia coli strain BL21 (DE3). The clones were analyzed by a wide-field upright fluorescence microscope (Stereo Discovery V8, Carl Zeiss) with blue light generated by the X-cite 120PC (mercury lamp) equipped with proper filter set. Then these mutants were analyzed by our home-made wide-field microscope to confirm their photo-switchable properties under 488-nm and 405-nm lasers. At last, 11 mutants with high fluorescent brightness and high on-off contrast were selected for this study.

**Plasmids**. The plasmids for bacterial expression of variants from Skylan-NS[17], rsFolder, and rsEGFP2[29] were provided by Pingyong Xu and Dominique Bourgeois, respectively. The members of the Skylan series are denoted as Skylan-XXX (X represents the α-amino acid at position 62, 157, 173 respectively). The numbering adopted for the RSFPs in this manuscript is provided in the Supplementary Information together with the sequences of the genes coding for these proteins (see section 1 in Supplementary Information).

**Protein production and purification**. The plasmids expressing proteins with an N-terminal hexahistidine tag were transformed in E. coli BL21 strain. Cells were grown in Terrific Broth (TB) at 37 ºC. The expression was induced at 30 ºC or 16 ºC by addition of isopropyl β-D-1-thio-galactopyranoside (IPTG) to a final concentration of 1 mM at OD(600) = 0.6. The cells were harvested after 16 h of expression and lysed by sonication in lysis buffer (50 mM PBS with 150 mM NaCl at pH 7.4, 5 mg/ml DNAse, 5 mM MgCl₂, and 1 mM phenylmethylsulfonyl fluoride (PMSF), and a cocktail of protease inhibitors (Sigma Aldrich; S8830)). After lysis, the mixture was incubated on ice for 2 h for DNA digestion. The insoluble material was removed by centrifugation and the supernatant was incubated overnight with Ni-NTA agarose beads (Thermo Fisher) at 4 ºC in a rotator-mixer. The protein-loaded Ni-NTA column was washed twice with 20 column volumes of N1 buffer (50 mM PBS, 300 mM NaCl, 30 mM imidazole, pH 7.4) and twice with N2 buffer (50 mM PBS, 150 mM NaCl, 10 mM imidazole, pH 7.4). The bound protein was eluted with N3 buffer (150 mM PBS pH 7.4, 300 mM imidazole). The protein fractions were eventually dialyzed with cassette Slide-A-Lyzer Dialysis Cassettes (Thermo Fisher) against 50 mM PBS, 150 mM NaCl pH 7.4.

**Production of RSFP-labeled Escherichia coli**. Escherichia coli cells from the TOP10 strain were transformed with the RSFP Plasmids by electroporation. The transformed E. coli cells were grown at 37 °C in LB broth. When the optical density at 600 nm reached 0.2, expression was induced by addition of isopropyl β-D-1-thio-galactopyranoside (IPTG) to a final concentration of 1 mM. After 4 hours of expression at 30 °C, 1 mL aliquots were taken and cells were centrifuged at 8000 rpm for 5 min. After centrifugation, the supernatant was removed and the E. coli cells were washed once with 1 mL of PBS (pH 7.4, 50 mM sodium phosphate, 150 mM NaCl) and then resuspended in 250 μL of PBS buffer.

**Preparation of the samples of RSFP-labeled Escherichia coli**. Bacteria aliquots (1 mL) from cells culture were centrifuged at 8000 rpm for 5 min. After centrifugation, the supernatant was removed and the E. coli cells were washed once with 1 mL of PBS (pH 7.4, 50 mM sodium phosphate,150 mM NaCl), and then resuspended in 250 μL of 4% Formalin in PBS. The cells were mixed gently and incubated for 3 h at room temperature.

Glass slides were washed with 1 M sodium hydroxide, then rinsed with water, ethanol, and eventually dried. They were subsequently incubated for 30 min, with 0.01% poly-L lysine, washed once with water, and then dried.

About 3 μL of fixed bacteria was dropped onto the poly-L lysine-coated glass slide, which was finally covered with a cover slip for microscopy observation.

**Spectroscopic measurements**. UV/Vis absorption spectra were recorded on a UV/Vis spectrophotometer (Cary 300 UV-Vis, Agilent Technologies, Santa Clara, CA) at 20 °C equipped with a Peltier 1 × 1 thermostatic cell holder (Agilent Technologies). Samples (45 μL) were contained in 0.3 cm × 0.3 cm (light path) quartz cuvettes (Hellma Optics, Jena, Germany). Fluorescence measurements were acquired on a LPS 220 spectrofluorometer (PTI, Monmouth Junction, NJ), equipped with a TLC50 cuvette holder (Quantum Northwest, Liberty Lake, WA) thermoregulated at 25 °C. In addition to Fig. 3a, b, the normalized absorption and emission spectra of the 22 RSFPs, which have been investigated in this study are provided in Supplementary Figure 13.

**Measurements of the rate constant associated with thermal relaxation after photoswitching**. In order to study the thermal relaxation after phoisomerization upon illumination at $\lambda_1 = 488$ nm, we used the fluorimeter to record the

fluorescence emission from 5 $\mu$M RSFP solutions upon applying a series of light pulses at 480 nm up to reach the photostationary state using a LED (M470-L4, Thorlabs, NJ; filtered at 480 ± 20 nm (FF480-40, Semrock, Rochester, NY)) separated by increasing delays in the dark (denoted "Illumination III" in subsection A.2.1 in Supplementary Information). Since we adopted a regime of low light intensity in which a two-state model is relevant to account for the RSFP photoswitching behavior (see section C in Supplementary Information), we could satisfactorily use an exponential fitting function to account for the decay of the fluorescence signal occurring during each light pulse and accordingly extract its value at the initial time of each light pulse (see subsection A.2.1 in Supplementary Information). We eventually retrieved the thermal relaxation rate constant $k_{21}^{\Delta}$ from applying a monoexponential fitting function (see Eq. (58) in Supplementary Information) to the dependence of the values of the fluorescence signal at the initial time of each light pulse on the delay (see Supplementary Figure 14– Supplementary Figure 16). The results are provided in Supplementary Table 3.

**Acquisition of the RSFP photoswitching information**. The configuration displayed in Supplementary Figure 1a has been used to make measurements on RSFP solutions. With this configuration, we can automatically record and analyze in a few minutes the fluorescence response from a RSFP solution exposed to homogeneous illumination with microseconds time resolution.

To facilitate automated analysis, we tailored the geometry of illumination in order to control the fluorescence evolution by the light- and thermally-driven reactions without any significant interference of diffusion over the whole range of light intensities (see subsection A.2.2 in Supplementary Information). The RSFP solution is sandwiched in a 80 $\mu$m-thick measurement cell, which is put on a 0.4 mm thick copper disk. This metal holder is mounted on an aluminum block thermostated at 25 ± 0.2 °C with two thermoelectric Peltier devices (CP 1.0-63-05L-RTV, Melcor, Trenton, NJ). The stage temperature is monitored with a TCS610 thermistor (Wavelength Electronics, Bozeman, MT) and the feedback loop is driven by a MPT10000 temperature controller (Wavelength Electronics, Bozeman, MT).

The dedicated setup integrates two stages enabling us to illuminate the samples with Light Emitting Diodes (LEDs) or modulated laser diodes. Overall these light sources equip the setup with light intensities covering five orders of magnitude (from 0.5 W cm$^{-2}$ to 50 kW cm$^{-2}$).

The LED illumination stage integrates a blue color LED (M470L4, Thorlabs, NJ) filtered at $\lambda_1$ = 480 ± 20 nm (FF01-480-40, Semrock, Rochester, NY) and a UV LED (M405L3, Thorlabs, NJ) filtered at 405 ± 20 nm (ET 405/20x, Chroma Technologies, Bellows Falls, VT) as light sources. The current driving each LED is regulated by a LED driver (DC 4104, Thorlabs, NJ) operated in the external control mode and driven by the two analog outputs of a ADC card (usb-1604hs-2ao, Measurement Computing, Norton, MA) used for data acquisition or a waveform generator. A light condenser (ACL25416U, Thorlabs, Newton, NJ, $f$ = 16 mm) is placed just after each LED to collimate light and both light beams are combined with a dichroic mirror (T425LPXR, Chroma Technologies, Bellows Falls, VT). A second lens ($f$ = 100 mm) is used to focus the light at the back focal plane of the objective after being reflected by the dichroic filter (FF506-Di-03, Semrock, Rochester, NY). Thanks to two consecutive beam splitters (30:70, EBP1, Thorlabs and 50:50, CCM1-BS013/M, Thorlabs) and a lens (AC254-150, $f$ = 150 mm), part of the fluorescence emission at 525 ± 15 nm (FF525-30, Semrock, Rochester, NY) is focused on a multipixel photocounter (C13366-1350GA, Hamamatsu Photonics K.K., Hamamatsu City, Japan) used to measure fluorescence signals and on a CMOS sensor of a camera (3060cp, iDS, Obersulm, Germany) used to control the alignment of the beams and image the illuminated sample. Triggering and synchronization of light illuminations and data acquisitions is controlled by a Labview 2010 program (National Instruments, Austin, TX).

The beams of the laser diodes at $\lambda_1$ = 488 nm (LBX serie, Oxxius, Lannion, France) and $\lambda_2$ = 405 nm (06-01 Serie, Cobolt, Solna, Sweden) are first enlarged with a beam expander comprising two achromatic lenses (AC254-100-A, Thorlabs, Newton, NJ, $f$ = 100 mm and AC254-050-A, Thorlabs, Newton, NJ $f$ = 50 mm). A second afocal system made of two lenses (LA1213-A, Thorlabs, NJ, $f$ = 50 mm and LA1289-A, Thorlabs, NJ $f$ = 30 mm) is inserted in the 405 nm lightpath to correct possible chromatic aberrations of the imaging objective and improve the illumination homogeneity. The enlarged beams are combined with a dichroic mirror (T425LPXR, Chroma, Bellows Falls, VT) before the central part of the beams is selected with an iris (SM1D12CZ, Thorlabs, NJ, US) and focused with an achromatic lens (AC254-200-A, $f$ = 200 mm, Thorlabs, NJ, US) on the rear focal plane of a 50X objective (MPLFLN, NA 0.8, Olympus Corporation, Tokyo, Japan) after passing through a dichroic filter (FF506-Di-03, Semrock, Rochester, NY). The iris is conjugated to the focal plan of the objective, so that the size of the illuminated area can be adjusted by tuning the aperture of the iris. Thanks to a 30:70 beam splitter, 70% of the collected fluorescence emission at 525 ± 15 nm (FF525-30, Semrock, Rochester, NY) is focused with a lens (AC254-50, Thorlabs, $f$ = 50 mm) on the aperture of a 100 $\mu$m pinhole mounted in front of a photomultiplier tube (H10492-013, Hamamatsu Photonics K.K., Hamamatsu City, Japan) used to measure fluorescence signals while 15% of the collected fluorescence emission is focused on a CMOS sensor of a camera (3060cp, iDS, Obersulm, Germany) used to image the illuminated sample with a lens (AC254-150, $f$ = 150 mm). Triggering and synchronization of light illuminations and data

acquisitions is controlled by a Labview 2010 program (National Instruments, Austin, TX). In this system, the laser sources are triggered by a waveform generator (33612 A, Keysight Technologies) either in analog (488 nm; accepting an arbitrary 0–5 V input signal) or in digital (405 nm) mode.

**Acquisition of the LIGHTNING images**. The second configuration of our dedicated optical setup was optimized to provide images of RSFP-labeled bacteria with a 20 $\mu$m-diameter circular field of view at 1 kHz frequency of image acquisition by fixing the illuminations at tailored light intensities (see Supplementary Figure 1b). The bacteria samples have been put on the metal holder, which is described above. LIGHTNING imaging requires excitation at rather high light intensities, which led us to only retain lasers for illumination in the imaging setup.

Fluorescence images above 507 nm (FF-01-507/LP, Semrock) were acquired with a 50X objective (MPFLN, NA 0.80, Olympus Corporation) and reconstructed onto the sensor of a sCMOS camera (Orca-Flash v3, Hamamatsu Photonics K.K., Hamamatsu City, Japan) with a $f$ = 150 mm tube lens (ACA254-150-A). Triggering of the camera acquisition was synchronized with the onset of the excitation light. Data acquisitions were controlled by HCImage (Hamamatsu Corporation).

**Statistics and reproducibility**. All the experiments reported in this manuscript have been reproduced at least twice.

**Reporting Summary**. Further information on research design is available in the Nature Research Reporting Summary linked to this article.

## Data availability

All data generated or analyzed during this study are included in this published article and its supplementary information file.

## Code availability

The data were collected with Labview 2010 (to acquire photochemical information) and HCImage 4.4.2.7 (Hamamatsu Corporation, for imaging). The data were processed and visualized with Igor Pro 8, Mathematica 12.3.1.0, Matlab 2019b, Python 3.6, and Inkscape 1.1. The code for processing the images of bacteria is available at https://doi.org/10.5281/zenodo.5684342.

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

## Acknowledgements

Virgile Adam, Ninon Zala, and Kyprianos Hadjidemetriou are acknowledged for their help in producing RSFPs. We also thank David Bensimon, Camille Chartier, Jacques Fattaccioli, Arnaud Gautier, Zoher Gueroui, Pierre Neveu, and Alison Tebo for their supports and fruitful comments. IBS acknowledges integration into the Interdisciplinary Research Institute of Grenoble (IRIG, CEA). **Funding**: This work was supported by the ANR (France BioImaging - ANR-10-INBS-04 (L.J.), Morphoscope2—ANR-11-EQPX-0029 (L.J.), IPGG—ANR-10-IDEX-0001-02 PSL (L.J.), ANR-10-LABX-31 (L.J.), ANR-17-CE11-0047-01 (D.B.), ANR-19-CE11-0005 (T.L.S.), and ANR-19-CE29-0003-01 (A.E.)), the Fondation de la Recherche Médicale (FRM DEI201512440) (L.J.), Sorbonne Université (Emergence 2019–2020, LIGHTNING) (L.J.), the Mission Interdisciplinarité du CNRS (T.L.S.), the European Innovation Council (EIC Pathfinder Open 2021 DREAM) (L.J.), the Strategic Priority Research Program of Chinese Academy of Sciences (XDB37040301) (P.X.), and the National Natural Science Foundation of China (21778069) (P.X.).

## Author contributions

Conceptualization, A.E., A.Le., T.L.S., and L.J.; Methodology, A.P.-T., A.E., A.Le., T.L.S., and L.J.; Softwares, A.P.-T., A.La., A.Le., and T.L.S.; Formal analysis, R.C., N.D., A.P.-T., A.La., D.K., A.Le., T.L.S, and L.J.; Investigation, R.C., A.P.-T., A.La., D.K., and T.L.S.; Resources, R.C., A.P.-T., A.La., R.Z., M.-A.P., M.Z., X.Z., P.X., N.D., D.B., A.E., A.Le., and T.L.S.; Writing - Original Draft, R.C., A.P.-T., A.La., P.X., A.Le., T.L.S., and L.J.; Writing-Review & Editing, R.C., A.P.-T., A.La., P.X., N.D., D.B., A.E., A.Le., T.L.S., and L.J.; Funding Acquisition, A.E., A.Le., T.L.S., and L.J.

## Competing interests

R.C., A.P.-T., A.E., A.Le., T.L.S., and L.J. have a patent related to this work. There are no other competing interests.
