## [Peer Review File · Nature Communications]

Extra kinetic dimensions for label discriminationREVIEWER COMMENTS

Reviewer #1 (Remarks to the Author):

The manuscript by Chouket and coworkers describes a method for discrimination between fluorophores exhibiting significantly overlapping spectra by taking advantage of temporal dynamics of photoswitching under various saturation levels. This work is, in essence, an extension of past work by the same group ("OPIOM", Ref. 13) where additional kinetic dimensions are used for the discrimination. The authors convincingly show that they can differentiate between 30 fluorescent proteins exhibiting partially overlapping spectra although they measure at a single emission wavelength.

While the concept of the work is not new (these ideas have also been somewhat explored by others, see for example Chen and Dickson JPCL 8, 733, 2017) the experimental demonstration is quite impressive.

The most major comment regarding this method relates to its robustness. Since, as stated in the manuscript, the authors "used a monoexponential function to fit to the part of the fluorescence evolution exhibiting the largest amplitude over the time window for signal acquisition in order to retrieve a single characteristic time", it is not clear how well this method would work in the presence of more significant interference than what is presented in the text. This could even be autofluorescence in samples which do not emit so strongly, or systematic shifts such as the ones which seem to appear in Fig. 4c (where clear changes in color within the same sample are observed in many of the images), especially in very dense regions of the parameter space. Moreover, for samples labeled with multiple stains which nominally emit in different colors, there could be leakage between different color channels.

If the authors convincingly address this issue I believe that this manuscript can be accepted for publication in Nature communications. From a technical standpoint, it is very clearly written.

Reviewer #2 (Remarks to the Author):

I am very familiar with Ref. 5, although I am not one of the authors of this article. I agree wholeheartedly with the contention that present fluorescence labeling is limited to observing only a few fluorophores in a sample because of the overlap of fluorescence spectra. This manuscript presents an exciting new way to overcome this problem by relying on changes in fluorescence caused by changes in the excitation intensity. I do recommend publication, but I think some cautions need to be added to make a more balanced view.

The use of real-time fluorescence imaging in biology really means imaging different tags at the same time using different channels to see the signal cross talk between targets without a time delay. If this statement is correct, then some qualifications are needed in the present version of the manuscript. The method presented seems to be based on sequential illuminations where there is a signal delay for different tags so I would claim that the lack of real-time signals may limit its use in biological applications for some cases. I think this needs to be mentioned and discussed.

Another matter deserving more attention is the question of resolution. Although the authors claim a good resolution, I'm not sure their resolution is sufficient considering the probe's changes in properties based on its local environment. It can be seen from Fig 4 that although the authors claimed 8 out of 16 probes can be distinguished, (3, 8) and (19, 22) may be hard to distinguish due to the dispersion in characteristic time shown for the same figure/probe (see 17 for an example: it seems like the same probes could show quite large differences in characteristic times due to environmental differences). A deeper, more critical discussion is needed on this matter.

Paris, 21st November 2021,

We would like to thank the reviewers for their comments and criticisms. Below, we address point by point their different concerns:

Reviewer 1

The manuscript by Chouket and coworkers describes a method for discrimination between fluorophores exhibiting significantly overlapping spectra by taking advantage of temporal dynamics of photoswitching under various saturation levels. This work is, in essence, an extension of past work by the same group ("OPIOM", Ref. 13) where additional kinetic dimensions are used for the discrimination. The authors convincingly show that they can differentiate between 30 fluorescent proteins exhibiting partially overlapping spectra although they measure at a single emission wavelength. While the concept of the work is not new (these ideas have also been somewhat explored by others, see for example Chen and Dickson JPCL 8, 733, 2017) the experimental demonstration is quite impressive.

We thank the reviewer for his positive comments. We have made precise the differences between LIGHTNING and other imaging protocols of dynamic contrast in the Discussion in MT as follows:

“First, the latter usually exploit a single temporal dimension ($n = 1$), so that distinguishing many RSFs requires a larger span of characteristic times and a larger time window for signal acquisition than LIGHTNING which exploits several ($n = 3$ or 4) time dimensions (see section A.3.1 in SI). Then, the protocols involving an oscillating light excitation based on matching the period of the excitation and a typical time scale of photoswitching dynamics necessitate a kinetic model of RSF photoswitching.^{13,20} In contrast, LIGHTNING can be implemented without specific information on the mechanism of RSF photoswitching. Finally, instead of proceeding by successive acquisitions at optimal frequency targeting individual RSFs, LIGHTNING simultaneously discriminates multiple RSFs by applying a single illumination sequence of lights at constant intensities common to all RSF labels, which considerably shortens image acquisition.”

The reference to the paper Chen and Dickson JPCL 8, 733, 2017 has been added.

The most major comment regarding this method relates to its robustness. since, as stated in the manuscript, the authors "used a monoexponential function to fit to the part of the fluorescence evolution exhibiting the largest amplitude over the time window for signal acquisition in order to retrieve a single characteristic time", it is not clear how well this method would work in the presence of more significant interference than what is presented in the text. This could even be autofluorescence in samples which do not emit so strongly, or

systematic shifts such as the ones which seem to appear in Fig. 4c (where clear changes in color within the same sample are observed in many of the images), especially in very dense regions of the parameter space. Moreover, for samples labeled with multiple stains which nominally emit in different colors, there could be leakage between different color channels. If the authors convincingly address this issue I believe that this manuscript can be accepted for publication in Nature communications. From a technical standpoint, it is very clearly written.

The evaluation of LIGHTNING robustness is indeed an important issue that we addressed from different aspects.

– We first addressed the dispersion of the kinetic fingerprints of individual RSFPs within a population. In solution, the RSFP environment is homogeneous which yields a small dispersion, possibly limited by the instrumental setup (see section D.1.3 in SI). In contrast, in a biological sample, the RSFP environment may introduce a dispersion of photoswitching kinetics, which we found to exceed the dispersion originating from the instrumental setup (see section D.2.2 in SI). Please see the following discussion in section D.2.2 of SI:

“The variances and the covariances of the kinetic fingerprint $\{l_{i1}, l_{i2}, l_{i3}\}$ associated with the characteristic times $\{\tau_{low,i}^I, \tau_{high,i}^I, \tau_{high,i}^{II}\}$ of RSFP i -labeled bacteria are given in Tab. S14. According to Tab. S14, the typical value $M = 0.08$ is assigned to the uncertainty on the decimal logarithm of the characteristic times. Then Eq.(S19) is used to derive $d_c = 0.28$ for $n = 3$. As displayed in Fig.S5, the standard deviations of the distributions of the decimal logarithm of light intensity at $\lambda_1 = 488$ nm and $\lambda_2 = 405$ nm have been found equal to 0.07 and 0.04, respectively, which is smaller than the standard deviations of the distributions of l_{ki} . Hence we concluded that the standard deviation of the distributions of l_{ki} is dominated by the intrinsic dispersion of the characteristic times within the bacteria population and not by the dispersion of the light intensities of the instrumental setup as it was observed in RSFP solutions.”

– We then addressed the question of the influence of the environment on the dispersion of the kinetic fingerprints in detail and have added section D.3 to the revised manuscript. Please see the answer to the point 3. of the second reviewer.

– In addition, the reviewer is concerned with the relevance of extracting a single characteristic time using a monoexponential fitting. Another data processing method already present in the original manuscript allowed us to extract a single characteristic time (the argument of the extremum of the spectrum of characteristic times) shown to be very close to the result of monoexponential fitting without any hypotheses on the complexity of fluorescence evolution (see Fig S56 and section B3 in SI). The section LIGHTNING data collection in MT has been modified as follows:

“The extracted characteristic time cannot be necessarily interpreted as a physical time of chemical kinetics. Nevertheless, we proposed a mechanism of RSFP photocycle (see Fig. 1b) and discuss the illumination for which the extracted characteristic time matches an intrinsic property of kinetics (see section C in SI).”

A chemical species has been added to the RSFP photocycle in order to better account for the experimental results (see section C in SI).

– Finally, the reviewer is concerned with significant interference and possible leakage between different color channels for samples labeled with multiple stains. LIGHTNING exploits fluorophores with similar emission spectra (see Fig. 3b in MT and S13b in SI) and leakage would only occur if an interfering fluorophore would be present in the sample and contribute to fluorescence evolution. The following remarks have been added to section B.3 of SI:

“It is worth noting that autofluorescence can be modeled by a constant term in the fluorescence evolution already present in Eqs. (S77,S79,S81). The spectrum of characteristic times and the characteristic time

deduced from a monoexponential fitting are independent of constants terms. Consequently, data processing eliminates autofluorescence. The quality of the results obviously depends on the noise and the amplitude of fluorescence variation must be at least as large as the noise level.”

The case of time varying interference was already partly addressed in section B.2.1 of SI and has been significantly developed in the new section D.4 of SI. In addition, the section Discussion of MT has been completed as follows:

“Retrieving a LIGHTNING kinetic fingerprint is robust even when the targeted RSF is not the only fluorescent species (see section B in SI). When the interfering fluorescence signal is constant (e.g. autofluorescence), it is intrinsically discarded from data processing. When the interfering signal evolves with a characteristic time sufficiently different from the target, it is ignored by using an appropriate time window, which plays the role of a kinetic filter, i.e. selects a targeted characteristic time. Eventually, we developed a protocol to eliminate a background of fluorescence evolution associated with a characteristic time close to the target one (see section D.4 in SI). Interestingly, this protocol will be relevant to extract the kinetic fingerprints of colocalized RSFs.”

Reviewer 2

I am very familiar with Ref. 5, although I am not one of the authors of this article. I agree wholeheartedly with the contention that present fluorescence labeling is limited to observing only a few fluorophores in a sample because of the overlap of fluorescence spectra. This manuscript presents an exciting new way to overcome this problem by relying on changes in fluorescence caused by changes in the excitation intensity. I do recommend publication, but I think some cautions need to be added to make a more balanced view.

We thank the reviewer for recognizing the critical issue addressed in the manuscript.

The use of real-time fluorescence imaging in biology really means imaging different tags at the same time using different channels to see the signal cross talk between targets without a time delay. If this statement is correct, then some qualifications are needed in the present version of the manuscript. The method presented seems to be based on sequential illuminations where there is a signal delay for different tags so I would claim that the lack of real-time signals may limit its use in biological applications for some cases. I think this needs to be mentioned and discussed.

LIGHTNING does not require a specific illumination sequence to image a given RSFP, which would introduce time delays for acquiring an image containing multiple RSFPs. In contrast, we identified a common illumination sequence leading to an image of all RSFPs present in the sample in less than 1 s (see section A.3.4 of SI), which is compatible with many biological applications. The discrimination of the different RSFPs is achieved using post-acquisition data processing. The section The LIGHTNING concept in MT has been modified according to:

“Applying a sequence of n multicolor illuminations at different light intensities yields a set of at least n non-redundant characteristic times $\{\tau_i\}$, which provides the LIGHTNING kinetic fingerprint (see Fig.1b). Importantly, a single illumination sequence can be tailored to deliver the LIGHTNING kinetic fingerprints of all non colocalized RSFs present in the sample.”

Another matter deserving more attention is the question of resolution. Although the authors claim a good resolution, I'm not sure their resolution is sufficient considering the probe's changes in properties based on its local environment. It can be seen from Fig 4 that although the authors claimed 8 out of 16 probes can be distinguished, (3, 8) and (19, 22) may be hard to distinguished due to the disperse in characteristic time shown for the same figure/probe (see 17 for an example: it seems like the same probes could show quite large

differences in characteristic times due to environmental differences). A suggest a deeper, more critical discussion is needed on this matter.

– We first performed additional experiments to check the sensitivity of the LIGHTNING kinetic fingerprint to the environment. Please see the new section D.3 in SI. The section LIGHTNING in action of MT has been increased by:

“Eventually, we studied the fluorescence response of two representative RSFPs **1** and **2** in three distinct environments – solution, fixed, and living labeled *Escherichia coli* bacteria – to various $\{I_{\text{low}}, II_{\text{low}}, I_{\text{high}}, II_{\text{high}}\}$ illuminations using an acquisition frequency of 100 kHz in the $[1\mu\text{s}-0.25\text{s}]$ time window (see section D.3 in SI). For each illumination regime, (i) the dependence of the characteristic times on the light intensities agrees with the prediction of the eight-state photoswitching mechanism displayed in Fig. 1b (see section C); (ii) the three distinct environments lead to similar characteristic times differing by a factor of about 1.2 for **1** and 1.4 for **2** in average, which is in line with the consistency of $\tau_{\text{low}}^{\text{l}}$ and $\tau_{\text{high}}^{\text{l}}$ in Figs. 3c and 4a (see section D.3 in SI). The changes in environment induce a variability of the characteristic times, which remains lower than the resolution limit of the kinetic fingerprint determined for a large number of acquisitions in a given environment.”

– We agree that the original Fig. 4c did not do justice to the work. The chosen colorimetric scale did not allow us to visually distinguish two RSFPs although correctly discriminated by the method. This choice led to spurious similarities of the colors, for example between **3** and **18**, and between **19** and **22**.

We introduced another criterion to discriminate fixed RSFP-labeled bacteria using their kinetic fingerprints, which have been processed again. This led us to select a slightly different subset of 9 discriminatable RSFP-labeled bacteria instead of 8. The results are shown in the revised Fig. 4c and the new Fig. 4d. Please see the new sections 1.8 and D.2 in SI. The section LIGHTNING in action of the MT has been modified as follows:

“The average values and standard deviations of the distributions of characteristic times were used to compute the resolution limit $d_c = 0.28$ and identify an optimized subset of 9 RSFP-expressing bacteria. They were also exploited to compute the maximum probability that a bacterium is identical to a tabulated RSFP-labeled bacterium, the identity of which was assigned to the bacterium (see sections 1.8 and D.2 in SI). Figure 4c illustrates the results and the confusion matrix shown in Fig. 4d gives the proportion of true and false positives. The accuracy defined as the ratio of the number of correctly identified bacteria and the total number of RSFP-labeled bacteria is equal to 0.93, establishing that LIGHTNING can reliably identify 9 RSFPs-labeled bacteria among 16.”

A false color in the revised Fig. 4c corresponds to a given predicted type of RSFPs-labeled bacteria.

REVIEWER COMMENTS

Reviewer #1 (Remarks to the Author):

I believe the authors have done excellent work in revising their manuscript, which can be practically accepted in its present form.

One small comment is that I did not understand the proposed extension to Raman microscopy (exactly what transients are they proposing to look at there?)

Reviewer #2 (Remarks to the Author):

The authors have considered seriously the questions I raised and have succeeded in answering them in a style that will be helpful to other readers. I recommend publication.

Paris, 13th December 2021,

We would like to thank the reviewers for their comments. Below, we address point by point their concerns:

Reviewer 1

I believe the authors have done excellent work in revising their manuscript, which can be practically accepted in its present form.

We thank the reviewer for his positive comments.

One small comment is that I did not understand the proposed extension to Raman microscopy (exactly what transients are they proposing to look at there?).

We agree with the reviewer that the sentence “LIGHTNING also promises further improvements and generalization to other spectroscopies (e.g. Raman microscopy)” was too compact to make explicit our proposal to adapt the LIGHTNING principle on other spectroscopies than fluorescence for retracing the time evolution of a reversibly photoswitchable label under illuminations. Hence, we revised this sentence and it is now written in the revised version of the manuscript “In this work, LIGHTNING has been implemented to successfully discriminate an unprecedented number of fluorescent proteins using simple tools. However, its principle could be transposed to other spectroscopies than fluorescence for label reporting under illumination (e.g. to further improve multiplexing in Raman microscopy).”

Reviewer 2

The authors have considered seriously the questions I raised and have succeeded in answering them in a style that will be helpful to other readers. I recommend publication.

We thank the reviewer for recommending publication of our manuscript.